# Social Isolation Activates Dormant Mammary Tumors, and Modifies Inflammatory and Mitochondrial Metabolic Pathways in the Rat Mammary Gland

**DOI:** 10.3390/cells12060961

**Published:** 2023-03-21

**Authors:** Fabia de Oliveira Andrade, Lu Jin, Robert Clarke, Imani Wood, MaryAnn Dutton, Chezaray Anjorin, Grace Rubin, Audrey Gao, Surojeet Sengupta, Kevin FitzGerald, Leena Hilakivi-Clarke

**Affiliations:** 1Department of Oncology, Georgetown University, Washington, DC 20057, USA; deoli038@umn.edu (F.d.O.A.);; 2Hormel Institute, University of Minnesota, 801 16th Ave NE, Austin, MN 55912, USA; 3Department of Psychiatry, Georgetown University, Washington, DC 20057, USA; 4Department of Medical Humanities, Creighton University, Omaha, NE 68178, USA

**Keywords:** social isolation, tamoxifen, breast cancer recurrence, Jaeumganghwa-tang, IL6/STAT3, oxidative phosphorylation, rat

## Abstract

Although multifactorial in origin, one of the most impactful consequences of social isolation is an increase in breast cancer mortality. How this happens is unknown, but many studies have shown that social isolation increases circulating inflammatory cytokines and impairs mitochondrial metabolism. Using a preclinical Sprague Dawley rat model of estrogen receptor-positive breast cancer, we investigated whether social isolation impairs the response to tamoxifen therapy and increases the risk of tumors emerging from dormancy, and thus their recurrence. We also studied which signaling pathways in the mammary glands may be affected by social isolation in tamoxifen treated rats, and whether an anti-inflammatory herbal mixture blocks the effects of social isolation. Social isolation increased the risk of dormant mammary tumor recurrence after tamoxifen therapy. The elevated recurrence risk was associated with changes in multiple signaling pathways including an upregulation of IL6/JAK/STAT3 signaling in the mammary glands and tumors and suppression of the mitochondrial oxidative phosphorylation (OXPHOS) pathway. In addition, social isolation increased the expression of receptor for advanced glycation end-products (RAGE), consistent with impaired insulin sensitivity and weight gain linked to social isolation. In socially isolated animals, the herbal product inhibited IL6/JAK/STAT3 signaling, upregulated OXPHOS signaling, suppressed the expression of RAGE ligands S100a8 and S100a9, and prevented the increase in recurrence of dormant mammary tumors. Increased breast cancer mortality among socially isolated survivors may be most effectively prevented by focusing on the period following the completion of hormone therapy using interventions that simultaneously target several different pathways including inflammatory and mitochondrial metabolism pathways.

## 1. Introduction

Social isolation, characterized by perceived loneliness or a lack of social contact, is a powerful predictor of increased all-cause mortality [1]. Socially isolated individuals are more likely to develop ischemic heart disease, suffer from stroke, and die from these diseases than socially integrated individuals [2]. Social isolation also increases the risk of developing type 2 diabetes [3], dementia [4], worsens neurological disease symptoms [5], and cancer mortality [6,7]. The biological changes induced by social isolation that cause an increase in mortality remain unknown. Among the causes of social isolation are being elderly, poor, being discriminated due to race, ethnicity, religion, or gender identity, or having been diagnosed with a life-threatening disease such as cancer. Former U.S. Surgeon General Vivek Murthy published a book in 2020, entitled *Together: The Healing Power of Connection in a Sometimes Lonely World* to highlight how loneliness is a public health concern [8]. COVID-19 further brought an unprecedented level of social isolation to human societies worldwide [9].

Many studies have investigated the link between social isolation and breast cancer. In a pooled analysis of 9267 breast cancer patients, 16–41% were identified as feeling socially isolated when assessed 6 months to 2 years after their cancer diagnosis [10]. Moreover, socially isolated breast cancer survivors had a 43% higher risk of recurrence and a 64% higher risk of breast cancer-specific mortality than socially integrated survivors [10]. Many other studies have reported similar findings [11,12]. To reduce the risk of recurrence and breast cancer mortality among socially isolated patients, it is critical to determine the mechanisms of these interactions and to identify effective therapies to prevent recurrence. It is unlikely that social isolation causes a single gene change in mammary cancer that explains tumor recurrence. Rather, via the hypothalamic pituitary axis and autonomic nervous system, social isolation probably influences many biological systems that then alter the tumor microenvironment and the tumor itself. It has been suggested that successful cancer therapies include both tumor specific treatments and treatments that correct changes in host generated metabolites or dysfunctional neuroendocrine and pro-inflammatory and immune system, which all promote tumor growth [13].

In humans, the most frequently reported biological change linked to loneliness and social isolation is an increase in the circulating inflammatory markers [14,15]. In animal models, social isolation has been reported to impair metabolism [16] and mitochondrial oxidative phosphorylation (OXPHOS) [17,18]. If these changes explain the effects of social isolation on breast cancer recurrence, interventions that reverse them could reduce breast cancer mortality. Previously, we found that Jaeumganghwa-tang (JGT), a mixture of 12 herbs commonly and safely used in Asian countries for a wide range of ailments, reduced IL6 expression in mammary tumors and increased sensitivity to tamoxifen therapy in vitro and in an animal model [19]. Other studies have reported the ability of JGT to inhibit inflammatory cytokines in human mast cells in vitro [20] and in vivo in mice [21]. JGT also inhibited the growth of HT1080 human fibrosarcoma cells, human gastric carcinoma AGS, and human prostate carcinoma PC-3 cells in vitro [22]. An additional benefit of a herbal mixture is that it might be more potent and less toxic than single agents in reducing inflammation because the combinations potentiate the efficacy of individual herbs and counteract the harmful side effects of each other [23]. As a staple of traditional Asian medicine, JGT can be obtained from producers who follow strict quality control requirements and guarantee that all individual herbs are within the official specifications.

Here, we investigated whether social isolation causes a resistance to tamoxifen therapy and/or causes responding tumors to re-emerge from dormancy. Single housing, which elicits anxiety and other fearful behaviors, is a well-established animal model of social isolation [24,25]. Different rodent models have been used to study the impact of social isolation on breast cancer risk. In these studies, social isolation increased mammary cancer risk in C3(1)/SV40 T-antigen (SV40Tag) mice [25], TgMMTVneu mice [26], and mammary carcinogen-treated mice [27,28]. In addition, aging rats housed singly developed more spontaneous mammary tumors than group-housed rats [24]. Furthermore, in the 4T1 syngeneic mouse mammary tumor model, social isolation significantly increased tumor growth [29] and cancer mortality [30]. However, no earlier studies have explored whether social isolation influences the effectiveness of hormone therapies against breast cancer.

We found that after tamoxifen therapy was completed, social isolation induced the regrowth of dormant mammary tumors, increasing the risk of local mammary cancer recurrence. RNA sequencing data from mammary glands identified two key changes in socially isolated rats: enrichment of inflammatory pathways including IL6/JAK/STAT3 and the suppression of the OXPHOS pathway. JGT reversed these changes and maintained the dormancy of tamoxifen responsive mammary tumors in socially isolated rats.

## 2. Methods

### 2.1. Animals

We used Sprague Dawley rats that are known to be responsive to tamoxifen therapy to investigate whether JGT modifies the effect of social isolation on tamoxifen response and local recurrence after tamoxifen therapy ended. Eighty Sprague Dawley rats were obtained from Envigo and arrived at 6 weeks of age at the Georgetown University Animal Facility located at the Department of Comparative Medicine. The rats were housed in groups of three per cage. All rats were fed a semi-purified AIN93G diet. The animals were housed in a temperature- and humidity-controlled room with a 12-h light–dark cycle. All animal procedures were approved by the Georgetown University Animal Care and Use Committee to ensure humane care.

### 2.2. Mammary Tumor Induction and Social Isolation

ER+ mammary tumors were induced by the administration of 10 mg of 7,12-dimethylbenz[a]anthracene (DMBA, Sigma, St. Louis, MO, USA) diluted in 1 mL of peanut oil by gavage when the rats were 50-days of age. Tumor development was checked weekly and when the first tumor became palpable per animal, rats were divided into two groups: those kept group-housed (3 animals per cage, *n* = 40), or those housed singly (social isolation, *n* = 40).

### 2.3. Tamoxifen Therapy and Administration of Jaeumganghwa-Tang (JGT)

When the first tumor per animal reached a diameter of ~11 mm, group-housed and socially isolated rats were divided into two additional groups. From this point forward, the experiment contained four groups: group-housed treated with tamoxifen (*n* = 19), group-housed treated with tamoxifen + JGT (*n* = 17), socially isolated treated with tamoxifen (*n* = 17), and socially isolated treated with tamoxifen + JGT (*n* = 18). Four group-housed and five socially isolated rats that never developed mammary tumors that reached 11 mm in diameter were not included in the study. Tamoxifen was added to the AIN93G diet at a concentration of 340 ppm tamoxifen citrate. JGT was administered via drinking water (500 mg/kg body weight). JGT was produced by Hanjung Pharmaceuticals (165-7 Sangseo-dong, Daedeok-gu, Daejeon, Korea) based on the formulation approved by the Korean Ministry of Food and Drug Safety (MFDS). This company manufactures JGT under the Good Manufacturing Practice (GMP) guidelines established by the MFDS. All individual herbs were within the specification of the Korean Pharmacocopia 11th edition, and the final quality control was established by the analysis of three index materials: berberine, glycyrrhizic acid, and paeoniflorin. In our study, JGT was in powder form and was used before the expiration date. The number was MJK701. The composition of the JGT is shown in Appendix A.

### 2.4. Monitoring Mammary Tumor Responses

Response to tamoxifen treatment was divided into four categories: (1) complete response (CR, tumor disappearance); (2) partial response (PR, tumor stopped growing and/or shrank); (3) de novo resistance (DNR, tumor continued to grow regardless of tamoxifen treatment); and (4) acquired resistance (AR, tumor appeared after initiation of tamoxifen treatment and continued growing). Tumor response data were analyzed using the Chi test [2]. When a CR tumor remained nonpalpable for 9 weeks, tamoxifen was removed from the diet. Nine weeks of rat life corresponds to approximately 5 years of human life. Rats that received JGT with tamoxifen continued to receive JGT after the tamoxifen treatment ended. Regrowth of dormant mammary tumors, that is, local recurrence, was monitored for 9 weeks after tamoxifen administration.

### 2.5. Mammary Gland and Tumor Harvesting

At the end of the tumor response monitoring period, all tumors and fourth mammary glands (if they were tumor-free) were harvested. Half of the samples per mammary gland and tumor were paraffin embedded for histopathological analysis, and the other half were processed for RNA and protein analysis and stored at −80 °C.

### 2.6. Tumor Pathologic Evaluation

Formalin-fixed mammary tumors were embedded in paraffin and cut into 5 µm sections. Hematoxylin and eosin (H&E)-stained sections were then used for histopathological evaluation, which that was conducted by a veterinary pathologist, Dr. Galli, at Georgetown University.

### 2.7. RNA-Sequencing

RNA from tumor-free mammary glands was extracted using the Qiagen RNeasy Mini Kit (Qiagen, Hilden, Germany) according to the manufacturer’s instructions. The RNA concentration and purity were analyzed using a NanoDrop 1000 spectrophotometer (Thermo Scientific, Waltham, MA, USA). RNA-Seq was performed by Genomics and Epigenomics Shared Resource at Georgetown University Medical Center. Paired-end, dual-indexed libraries for RNA-Seq were constructed from 500 ng total RNA using the TruSeq Stranded Total RNA Kit (Illumina, San Diego, CA, USA) according to the manufacturer’s instructions. Briefly, coding RNA and multiple forms of non-coding RNAs were captured using bead-based cytoplasmic and mitochondrial rRNA depletion, cDNA synthesis, and PCR. The resulting sequencing libraries were assessed for quality using a BioAnalyzer 2100 with a DNA 1000 Kit (Agilent, Santa Clara, CA, USA) and quantified via fluorometry using the Qubit 4.0 (ThermoFisher). Libraries were sequenced on the NextSeq 550 system (Illumina) using the High Output Kit v2.5 (150 cycles) with a paired-end 75 bp read mode to an average depth of 50 M reads per sample. We used FastQC to check the raw data quality, lower quality reads, and trimmed Illumina adapter sequences. Gene expression levels were generated from Rsem [31] in combination with Bowtie2 and rat reference sequences (rn6). Differential expression analysis was performed using the DESeq2 package [32] in R, and FDR <0.1 as the cutoff point. A heat map was created for each set of filtered genes. Further functional analysis was performed using PANTHER v15.0 [33].

### 2.8. RNA-Seq Data Analysis

All raw data were passed through a FastQC quality check. Adapter trimming was first performed on raw data using cutadapt (v2.9). The reference genome was downloaded from Ensembl (Mus musculus release 99), and the reference genome index was built using Bowtie2 (v2.4.1). Paired-end trimmed read alignment and raw read count calculations were conducted using RSEM (v1.3.1). Statistical analyses were performed using the DESeq2 package (v1.26) in R software (v3.6). Genes with a *p*-value < 0.05 were considered as differentially expressed and used as the input for gene set enrichment analysis (GSEA) (v3.0, Broad Institute).

### 2.9. Knowledge-Guided Differential Dependency Network (KDDN) Analysis

The network model was created using the KDDN app (v1.1.0) in Cytoscape (v3.6.0) with an automatic optimal parameter. PPI information was obtained from the search results produced by the STITCH database (v5.0).

### 2.10. Quantitative Real-Time Polymerase Chain Reaction

RNA from mammary tumors was extracted as described for the mammary glands. Two micrograms of RNA from the mammary tumors and mammary glands was converted to cDNA using a High-Capacity cDNA Reverse Transcription Kit (Applied Biosystems, Waltham, MA, USA) in a PTC-100 thermal cycler (Bio-Rad, Hercules, CA, USA). cDNA at 5 ng/µL mixed with BrightGreen 2X qPCR MasterMix-ROX (abm, Inc., Richmond, Canada), and gene-specific forward and reverse primers were used for real-time PCR. The PCR was carried out using the QuantStudio 12 K Flex Real-Time PCR System (Applied Biosystems). The relative standard curve method was used to calculate the expression levels of the gene targets normalized to the housekeeping gene *Hprt1* in rat tissue. Primers used in qPCR analysis were designed using the IDT tool (Integrated DNA Technologies Coralville, IA, USA, primer sequence found in Appendix A).

### 2.11. Protein Extraction and Western Blotting

Proteins were isolated from mammary glands, mammary tumors, and the brain using Pierce RIPA lysis buffer (Thermo Scientific), supplemented with Mini Complete Protease Inhibitor (Roche, Mannheim, Germany) and PhosSTOP phosphatase inhibitor (Roche). A BCA Protein Assay Kit (Thermo Scientific) was used to measure the protein concentration according to the manufacturer’s protocol. Protein extracts were separated on a 4–12% gradient denaturing polyacrylamide gel (SDS-PAGE) and transferred to a nitrocellulose membranes using the Invitrogen iBlot 7-min Blotting System. Unspecific reactions were blocked with 5% non-fat dry milk diluted in Tris buffered saline  +  Tween 20 (TBST) for 1 h at room temperature. Membranes were then incubated overnight at 4 °C with specific primary antibodies (1:1000): receptor for advanced glycation end-products (RAGE) (37,647, Abcam, Cambridge, MA, USA) and pSTAT3 (9145, Cell Signaling Technology, Danvers, MA, USA). After washing with TBST, the membranes were incubated with the secondary antibody at room temperature for 1 h. Membranes were developed using SuperSignal chemiluminescent HRP substrate (Thermo Scientific) and the signals were captured using the Amersham imaging system (GE, Boston, MA, USA). After the development of pSTAT3 and RAGE, membranes were incubated with Restore^TM^ Western Blot Stripping Buffer (ThermoFisher Scientific) for 15 min, blocked with 5% nonfat dry milk in TBST for 1 h, and incubated overnight with total STAT3 (1:1000, #9139, Cell Signaling) and β-actin (1:1000, #8457, Cell Signaling), respectively. Stripping was confirmed by developing membranes using an Amersham imaging system. Protein levels were determined by band intensity using Quantity One software (Bio-Rad), and pSTAT3 was normalized to total STAT3 and RAGE normalized to β-actin expression.

### 2.12. Lactate Dehydrogenase (LDH) Activity and Lactate Level Assays

LDH activity and lactate levels were measured using a LDH Assay Kit (Abcam; ab102526) and L-lactate Assay Kit (Abcam; ab65331), respectively, according to the manufacturer’s instructions. The mammary gland was snap-frozen in liquid nitrogen and stored at −80 °C. The tissue was crushed in liquid nitrogen, and 100 mg and 30 mg were used for the LDH and lactate assays, respectively. The optical density of the samples was measured at 450 nm at the end of the reaction for the lactate assay and every 3 min for 90 min for LDH activity using a Synergy H1 microplate reader (Biotek, Winooski, VT, USA). Lactate levels and LDH activity were normalized to the protein levels measured using a BCA Protein Assay Kit (Thermo Scientific).

### 2.13. Statistical Analysis

The Chi-square test was used to assess the tumor response to tamoxifen, tumor recurrence, and tumor histopathology. Differences in tumor burden were assessed using two-way repeated measures ANOVA followed by the Tukey post hoc test. The *t*-test was used to assess the differences in the gene and protein expression between the two groups. Differential gene and protein expression among tumors from different groups was assessed by three-way ANOVA or two-way ANOVA followed by the Holm–Sidak post hoc test. Differences between groups were considered statistically significant when the *p* values were ≤0.05.

## 3. Results

### 3.1. Social Isolation Does Not Modify Tamoxifen Responsiveness during Therapy

ER+ mammary tumors were induced in Sprague Dawley rats by administering DMBA via oral gavage. DMBA is a polycyclic aromatic hydrocarbon (PAH), and PAHs in the human environment are linked to an increased breast cancer risk [34]. When a rat developed a palpable mammary tumor, we divided animals into two groups: they were either socially isolated by single housing or allowed to remain group-housed (GH). Initiating social isolation (SI) at this time point creates a model that mimics patients who feel socially isolated because of the stress of being diagnosed with breast cancer. Furthermore, findings from a previous study suggested that SI promotes breast cancer growth only when implemented after tumors are already present [35]. The experimental design is illustrated in Figure 1. When the first mammary tumor reached a size of approximately 11 mm in diameter, the SI and GH rats were divided into two additional groups: those that were treated with tamoxifen and those that were treated with tamoxifen + JGT. Tamoxifen was administered at a concentration of 340 ppm, which is relevant for human tamoxifen exposure levels via the AIN93G diet, as previously described [36]. JGT at a dose of 500 mg/kg was administered via drinking water as described previously [19]. This dose corresponds to 81 mg/kg when converted to a human exposure equivalent [37] and is less than humans can be safely exposed to over extended time periods [38].

At least half of all DMBA tumors in Sprague Dawley rats respond to tamoxifen [36,39]. In the present study, 31% (*n* = 18) of the 58 tumors that developed during the tumor monitoring period in GH rats exhibited a complete response, and 24% (*n* = 14) a partial response. Social isolation non-significantly reduced the rate of complete responses to 21% (*n* = 8 of a total of 39 tumors), but the rate of partial responses slightly increased to 28% (*n* = 11; ns) (Figure 2A). The proportion of de novo resistant tumors (tumor never responded) was 16% (*n* = 9) in the GH and 15% (*n* = 6) in the SI group. Of all the tumors detected during tamoxifen therapy, 29% (*n* = 17) of GH rats appeared after tamoxifen therapy started and 36% (*n* = 14) in the SI group; these tumors were considered to represent acquired resistance. None of the differences were statistically significant.

### 3.2. JGT Increases Responsiveness to Tamoxifen

As we have previously reported [19], giving rats JGT via drinking water significantly increased the rate of complete responses to tamoxifen in GH rats from 31% to 52% (*n* = 23 of a total of 44 tumors, *p* = 0.004) (Figure 2A). Similar results were observed in the SI rats, with the rate of complete responses increasing from 21% to 36% (*n* = 16 of a total of 45 tumors, *p* = 0.03).

### 3.3. Social Isolation Increases the Risk of the Regrowth of Dormant Mammary Tumors and Local Recurrence in Rats, and JGT Prevents this Increase 

Tamoxifen treatment was stopped 9 weeks after complete response, a timeframe representing a sustained response to this intervention. Social isolation significantly increased the risk that the responding tumors would emerge from dormancy and recur. Recurrence was defined as a tumor that regrew to at least 11 mm in diameter at the same location where the completely responding tumor was initially located. Figure 2B shows that among the GH animals, 45% of responding tumors recurred, whereas the rate of recurrence increased to 75% (*p* < 0.001) in the SI animals. The latency to tumor recurrence did not differ between the groups (Appendix A). As shown in Figure 2C, the tumor burden (sum of the area of all detected tumors per animal) after tamoxifen therapy was also significantly higher in SI rats than in GH rats. These findings indicate that the risk of regrowth of dormant mammary tumors is significantly greater in SI animals than in GH animals. In breast cancer patient populations, an increased rate of recurrence is associated with reduced overall survival, since recurring cancers are less responsive to therapies, and distant recurrence is generally fatal. JGT treatment continued after tamoxifen was removed in animals that had received a combination of tamoxifen and JGT. Continued JGT treatment prevented the increase in regrowth of dormant tumors and local recurrence in SI animals (Figure 2B). The percentage of recurrence in the SI rats decreased 3-fold from 75% without JGT to 22% with JGT (*p* < 0.001). JGT did not affect the incidence of local recurrence in GH rats (Figure 2B).

### 3.4. Tamoxifen and JGT Modify Tumor Histopathology

Most DMBA-induced mammary tumors in Sprague Dawley rats are malignant adenocarcinomas [36,39], as was also observed in this study (Appendix A). Other malignant mammary tumors detected in tamoxifen-treated rats were squamous carcinomas, adenosquamous carcinomas, and lipid-rich mammary carcinomas (Appendix A). Tamoxifen can increase the ratio of benign to malignant DMBA tumors [39], likely reflecting, in part, its established cancer-preventive activities in humans [40]. JGT further increased the proportion of benign tumors in the SI rats from 26% to 54% (*p* < 0.001; Figure 2D).

### 3.5. Social Isolation and JGT Modify IL6/JAK/STAT3 and Oxidative Phosphorylation Signaling in Mammary Glands and Tumors

We used RNA-Seq analysis to determine which signaling pathways were altered by social isolation and JGT treatment. We considered whether comparisons should be performed in mammary glands or tumors. If comparisons are made in tumors, they would have to occur between partially recurring or resistant tumors, and consequently, the data might be masked by tumor tamoxifen responsiveness rather than differences between GH and SI rats. In our earlier studies, both resistant and recurring tumors were associated with immunosuppression; therefore, differences between the control and experimental groups were seldom observed. Given the chemopreventive activities of tamoxifen, it is likely that relevant events also occur in normal, but carcinogen-exposed mammary glands. Hence, the fourth abdominal mammary gland was obtained from GH or SI rats for RNA-Seq.

#### 3.5.1. Effects of Social Isolation

Since we observed a difference in the proportion of rats with recurrent tumors after tamoxifen treatment, which was significantly higher in SI than GH rats, we used mammary glands obtained after tamoxifen therapy to determine which genes were significantly altered by social isolation. We identified 674 differentially expressed genes using the cutoff criteria of *p* < 0.05 and a fold-change ≥1.5 (Appendix A). Genes with an expression value of 0 in four or more samples were excluded. Gene Ontology (GO) analysis indicated that the main alterations in the mammary glands of GH and SI animals involved cell proliferation and cell metabolism (Appendix A).

We then performed gene set enrichment analysis (GSEA) to identify genes that may act together. The results from the GSEA analysis were used to identify the differentially expressed ‘Cancer Hallmark’ gene sets and KEGG pathways. The top Cancer Hallmark pathways enriched in the SI rats, compared with the group-housed rats, included IFNα, IFNγ, and inflammatory responses, and IL6/JAK/STAT3 and TNFα signaling via NFκB (Figure 3A and Figure 4A). The top Cancer Hallmark pathways suppressed in the SI were oxidative phosphorylation (OXPHOS), MYC targets VI and V2, E2F targets, and the G2M checkpoint (Figure 3A and Figure 4A). Many of these pathways are linked to mitochondrial metabolism and cell proliferation, reflecting the consistency between the results of GO and Cancer Hallmark pathway analyses. The results also indicated that the functions identified by GO analysis were disrupted rather than increased in the SI rats. Although MYC and E2F are often oncogenic, their inhibition also reflects mitochondrial dysfunction [41].

KEGG pathway analysis confirmed that SI upregulated the inflammatory pathways. Of the top 10 enriched pathways in the mammary glands of SI rats, six were inflammatory or immune cell signaling pathways (Figure 4B). The second top pathway that was inhibited in SI rats, compared with GH rats, in the KEGG analysis, was the OXPHOS pathway (Figure 4B). Other suppressed KEGG pathways in SI rats indicated ribosomal inactivation, impaired mismatch and nucleotide excision repair, DNA replication, and the tricarboxylic acid (TCA) cycle.

#### 3.5.2. Effects of JGT on Socially Isolated Rats after Tamoxifen Treatment

Since JGT prevented the increase in local recurrence in SI rats, we evaluated the differences in gene expression in the mammary glands of SI rats that either continued to receive JGT after tamoxifen treatment or that never received JGT. Using the same criteria as described above, 349 differentially expressed genes were identified (Appendix A). The top pathways identified in GO analysis were related to antibodies and immune responses (Appendix A).

Analysis using the ‘Cancer Hallmark’ gene set indicated that the top pathways suppressed by JGT in SI rats were IFNα and IFNγ, and inflammatory responses and IL6/JAK/STAT3 signaling (Figure 3D and Figure 4C), that is, the same pathways that were activated in SI rats compared with GH rats. In the KEGG pathway analysis, nine of the top 10 inhibited pathways in SI rats treated with JGT were the inflammatory and immune cell signaling pathways (Figure 4D). JGT caused an enrichment of OXPHOS in SI rats in both the Cancer Hallmark pathway analysis (Figure 3D and Figure 4C) and KEGG pathway analysis (Figure 4D). The TCA cycle pathway was also upregulated by JGT in the KEGG analysis of SI rats.

#### 3.5.3. Genes Altered by Social Isolation and Reversed JGT in the IL6/JAK/STAT3 and OXPHOS Pathways

We investigated the common genes that contributed to the change in the IL6/JAK/STAT3 and OXPHOS Cancer Hallmark pathways between the GH and SI rats and were reversed by JGT in the SI rats.

##### IL6/JAK/STAT3 Pathway

In the IL6/JAK/STAT3 pathway, 17 genes shown in Appendix A were upregulated in SI rats and then reduced by JGT. The specific functions of the 17 genes are listed in Appendix A. Among these genes are CD14, CSF2, and CXCL10, which are linked to COVID-19 related cytokine storm (CD14 [42] and CXCL10 [43]) and acute respiratory syndrome (ARDS) (CSF2/GM-CSF [44]).

###### OXPHOS Pathway

Appendix A shows the genes that contributed to the suppression of the OXPHOS Hallmark pathway in SI rats, and that JGT reversed. SI suppressed several genes linked to the TCA cycle and its activation as well as genes involved with mitochondrial electron transport chains. JGT upregulated the expression of the OXPHOS pathway genes in SI rats.

Among the inhibited OXPHOS pathway genes in SI rats was MPC1, which transports pyruvate into the mitochondria, and the mitochondrial pyruvate dehydrogenase complex genes (PDHB and PDHX) that convert pyruvate to acetyl CoA (Appendix A). Because impaired OXPHOS may occur when less pyruvate is provided to the mitochondria and more is converted to lactate, we measured the lactate dehydrogenase (LDH, an enzyme that converts pyruvate to lactate) and lactate levels. However, as illustrated in Appendix A, neither the LDH nor lactate levels were altered in the mammary glands of SI rats compared with GH rats. Since we used mammary glands rather than tumors in the RNA–Seq analysis and all animals had completed a long tamoxifen treatment, it is possible that different findings would have been obtained in non-tamoxifen treated mammary tumors, which were not available for this study.

#### 3.5.4. Effects of JGT on the Tamoxifen-Treated Group-Housed Rats

Finally, we determined whether JGT affects similar pathways in tamoxifen treated GH rats than in post-tamoxifen SI rats. Although JGT did not reduce the rate of local mammary tumor recurrences in GH rats, it improved their responsiveness to tamoxifen. Using the cutoff criteria noted above, we identified a total of 352 candidate genes that were significantly differentially expressed in the mammary glands of tamoxifen-treated GH rats compared with GH rats treated with tamoxifen + JGT (Appendix A). GO analysis implicated ‘opioid receptor binding’, ‘immune receptor activity’, and ‘cytokine binding’ as differentially activated GO molecular functions by JGT. ‘Negative regulation of IL6 production’, ‘inflammatory response’, and ‘cytokine production’ were among the altered GO biological processes (Appendix A). Although GO analysis results for JGT were different in post-tamoxifen SI rats and tamoxifen-treated GH rats, it was common in both analyses that the immune related functions were altered.

In the Cancer Hallmark pathway analysis, IFNα, IFNγ, inflammatory responses, and IL6/JAK/STAT3 signaling were suppressed significantly by JGT in tamoxifen-treated GH rats (Figure 3G and Figure 4E). KEGG pathway analysis indicated that among the top 10 inhibited pathways in GH rats treated with JGT, six were cytokine or other immune cell signaling pathways (Figure 4F). Furthermore, both in the Cancer Hallmarks and KEGG pathways, JGT enriched the OXPHOS pathway (Figure 3G and Figure 4E,F). Other Cancer Hallmark enriched pathways in JGT-treated GH rats were Myc targets V1, mTORC1 signaling, adipogenesis, and cholesterol homeostasis. Each of these pathways is linked to mitochondrial function, and their upregulation by JGT may indicate that this herb mix improved the mitochondrial metabolism. Indeed, JGT had similar effects on many Cancer Hallmarks and KEGG pathways in tamoxifen-treated GH rats and post-tamoxifen SI rats.

#### 3.5.5. Knowledge-Fused Differential Dependency Network (KDDN) Analysis

To better understand the connections among differentially expressed genes in the Cancer Hallmark pathway analysis, we performed KDDN analysis to identify novel connections induced by SI or JGT in the IL6/JAK/STAT3 and OXPHOS pathways. KDDN discovers unique signaling connections (edges) between genes (nodes) that are present only in GH or SI rats [45]. Hub genes, represented as nodes with multiple edges, are particularly important.

The unique edges in the IL6/JAK/STAT pathway that were present (green) or lost (red) in the KDDN analysis in SI rats when compared with GH rats or in SI rats treated with JGT are shown in Figure 3B,E, respectively. We observed that in the IL6/JAK/STAT3 pathway, connections from the Stat5a node to Ep300, Bcl2, and Socs1 were lost in the SI rats, but present in GH rats (Figure 3B). Ep300 functions as a histone acetyltransferase that regulates transcription via chromatin remodeling and can activate genes by suppressing histone deacetylase 1 (HDAC1) [46]. STAT5 binds to SOCS1 to provide feedback for the regulation of CD8^+^ T cells [47]. BCL2, in turn, can regulate immune cell survival by inhibiting apoptosis. The KDDN analysis indicates that these regulatory mechanisms were lost in rats with SI.

JGT-treated SI rats regained the connection between Stat5a and Ep300 but lost the connection between Stat5a and Ptpn2 (Figure 3E). Since PTPN2 promotes FoxP3/Treg stability [48], the KDDN results suggest that JGT may inhibit immunosuppressive Foxp3 cells in the tumor microenvironment.

Connections in the OXPHOS pathway (Figure 3C) included the hub gene Mrps30, which when downregulated, suppresses OXPHOS to promote breast cancer growth [49]. In the SI rats that exhibited increased risk of breast cancer recurrence, this gene was downregulated and had lost its connection to Ndufs3, Ndufa4 (both involved in mitochondrial membrane respiratory chain), and Vdac2 (pathway for metabolite diffusion across the mitochondrial outer membrane), and gained a connection to Suclg1 (TCA cycle) and Atp6v0c (enzyme transporter that acidifies intracellular compartments in eukaryotic cells). When SI rats were treated with JGT, Mrps30 gained a connection with Ndufc2 and lost a connection with Atp5fd (a member of the electron transfer complex V) (Figure 3F).

Most novel connections in the KDDN analysis were detected in GH rats treated with tamoxifen + JGT compared with those treated with tamoxifen alone. Unique edges in the IL6/JAK/STAT and OXPHOS Hallmark pathways in JGT-treated GH rats are shown in Figure 3H,I. For example, CMPK2 is a mitochondrial nucleotide kinase that supplies deoxyribonucleotides for mitochondrial DNA (mtDNA) synthesis to activate the Nod-like receptor protein 3 (NLRP3) inflammasome complex. This complex upregulates IL1β to induce inflammation [50]. As shown in Figure 3H, the connection between the two hubs, CMPK2 and IL1β, was lost in the JGT-treated rats. In JGT-treated animals, IL1β was associated with CSFR3 (involved in granulocyte differentiation) and IL10RA (anti-inflammatory/immunosuppression). These findings indicate that JGT induced a new, anti-inflammatory connection to replace inflammatory gene connection in the mammary gland of tamoxifen-treated GH rats.

KDDN analysis of the OXPHOS pathway indicates that gene interactions that promote the use of amino acids to generate ATP via the TCA cycle in GH rats (from Cox15 to Aldh6a1) are replaced by those that promote improved mitochondrial respiration (from Cox15 to Cox5a, Ndufc1, and Mrps30) in the JGT-treated GH rats (Figure 3I). In addition, JGT promoted the conversion of Acc2 to Glud1. Glud1 converts glutamate to α-ketoglutarate to activate the mitochondrial electron transport chain that generates the most energy to cells. In rats only treated with tamoxifen, Acc2 connects to Id2, which catalyzes the oxidative carboxylation of isocitrate to 2-oxoglutarate in the TCA cycle and to Htra2. Htra2 activates the cellular stress response. The new connections created by JGT treatment might explain how this herb mix upregulates the OXPHOS signaling pathway.

### 3.6. Verification of Differential Gene Expression

#### 3.6.1. Inflammatory Genes

While no single gene alone likely explains the complex effects of social isolation on breast cancer mortality, it is customary to verify the validity of key results from RNA-Seq and other omics analyses. We used RT-qPCR to validate the differential expression of the five genes most strongly linked to inflammation and tumor immune responses: chemokine *Ccl7*, cytokine *Csf2/GM-CSF*, cytokine receptors *Il4r* and *Il18r1*, and inhibitor of antitumor immunity through antigen presentation *Lilrb3*. The selected genes exhibited the minimum within-group variation in the RNA-Seq analysis. Consistent with the RNA-Seq data, *Csf2*, *Ilr4*, *Il18r1*, and *Lilrb3* mRNAs were significantly upregulated in the mammary glands of SI rats compared with GH rats (Figure 5A–D). The modest increase in the expression of *Ccl7* did not reach statistical significance (Figure 5E).

We then determined whether any of the gene expression changes detected between the GH and SI rats after tamoxifen treatment were reversed in SI rats supplemented with JGT. Of the five genes that were significantly or non-significantly upregulated in the SI rats, compared with the GH rats, JGT significantly reduced the expression of *Il18r1* and non-significantly reduced *Lilrb3* expression in the SI rats (Figure 5C,D).

#### 3.6.2. OXPHOS Genes

We measured the expression of the eight OXPHOS pathway genes that were downregulated in the SI rats when compared with GH rats in the RNA-Seq analysis. Specifically, cytochrome c-1 (*Cyc1*) is a member of the mitochondrial electron transport chain complex III, fructose 1,6 bisphosphate-2 (*Fbp2*) catalyzes the conversion of fructose-1,6BP to fructose-6P in the glycolysis to feed TCA and ultimately OXPHOS, isocitrate dehydrogenase (NAD(+))3 is the non-catalytic subunit gamma (*Idh3g*) that converts isocitrate to α-ketoglutarate in the TCA cycle, malate dehydrogenase 1 (*Mdh1*) converts malate to oxaloacetate in the TCA cycle, and NADH:ubiquinone oxidoreductase subunit A9 (*Ndufa9*), NADH:ubiquinone oxidoreductase subunit AB1 (*Ndufab1*), and NADH:ubiquinone oxidoreductase subunit C1 (*Ndufc1*) are all part of complex I of the electron transport chain. Succinate dehydroxygenase complex iron sulfur subunit B (*Sdhb*) is a member of complex II of the electron transport chain. *Fbp2*, *Idh3g*, *Ndufab1*, and *Ndufc1* were significantly lower in SI rats than in GH rats (Figure 6A–D). The expression of the other genes was reduced, but not significantly (Figure 6E–H).

Supplementing SI rats with JGT significantly upregulated the expression of *Cyc1*, *Idh3g*, *Ndfc1*, *Mdh1*, and *Sdhb*, and tended to upregulate *Ndufab1*. The inhibition of *Fbp2* and *Ndufa9* in SI rats was not reversed by JGT (Figure 6A,G). Although we did not confirm significant differences in all the genes that were among the altered genes in the IL6/JAK/STAT3 and OXPHOS signaling pathways in RNA-Seq analysis, non-significant changes likely contribute to the overall function of the two signaling pathways.

#### 3.6.3. Effect of JGT on Tamoxifen-Treated GH Rats

As implied by the RNA-Seq data, we investigated whether in GH rats during tamoxifen treatment, JGT suppressed the chemokines *Ccl12*, *Csf3r* (receptor for CSF3), *Mcemp1* (transmembrane protein possibly regulating mast cell differentiation and immune responses), or *s1008a* and *s1009a* (ligands for RAGE). *Mcemp1*, *S100a8,* and *s100a9* were significantly downregulated by JGT (Figure 7A–C); the expression of *Ccl12* or *Csf3r* was not altered (Figure 7D,E). These data suggest that RAGE signaling may be inhibited by JGT in tamoxifen-treated animals.

#### 3.6.4. IL6 and STAT3 Expression in Mammary Tumors

We also determined the changes in the IL6 and STAT3 levels in mammary tumors. Since there were insufficient numbers of tumors available for the analysis after tamoxifen therapy was completed, tumors were obtained from animals during tamoxifen treatment. In the histopathological analysis, some of the tumors were benign and some were malignant, as tamoxifen increases the proportion of benign DMBA tumors. Social isolation increased the *Il6* mRNA levels (Figure 8A) and pSTAT3 protein levels (Figure 8B) in benign mammary tumors. No differences were observed in malignant tumors, but the *Il6* levels were higher in malignant tumors than in benign tumors. JGT reduced the *Il6* and pSTAT3 expression in mammary tumors; this reduction was observed in malignant tumors for *Il6* and in benign tumors for pSTAT3.

### 3.7. Social Isolation Upregulates Receptors for RAGE in Mammary Glands and Tumors

As RAGE ligands were suppressed by JGT, we determined whether RAGE expression was altered in SI rats. RAGE is an inflammatory receptor that activates NF-κB, resulting in the production of proinflammatory cytokines IL1 and IL6 [51]. RAGE mRNA and protein levels were significantly higher in SI rats than in GH rats in both the mammary glands and the brain (Figure 9A). In the malignant mammary tumors during tamoxifen treatment, *Rage* mRNA expression was higher in the SI rats than in the GH rats (Figure 9B). Paradoxically, benign tumors in the GH rats exhibited higher *Rage* mRNA levels than malignant tumors. JGT did not affect the RAGE levels (Figure 9B).

We then determined whether the expression of the RAGE ligands *S1008a* or *S1009a* was altered in the benign or malignant mammary tumors between GH and SI animals during tamoxifen therapy (Figure 9C,D). However, JGT suppressed the expression of both *S1008a* and *S1009a* (Figure 9C,D), which is consistent with the data from the mammary glands (Figure 7B,C).

## 4. Discussion

In the preclinical setting, we found that social isolation did not affect the responsiveness to tamoxifen treatment. However, after tamoxifen therapy ended, the risk of local mammary tumor recurrence increased by 60% in SI rats compared in GH rats. Local recurrence in our study refers to a mammary tumor that responded to tamoxifen and consequently could no longer be palpated, but after tamoxifen treatment ended, the tumor reemerged and started to grow. In humans, this pattern of recurrence is often described as reflecting ‘dormancy’, which is a key feature of ER+ breast cancer and a major challenge in the eradication of this disease. If translatable to humans, the preclinical findings reported here suggest that the increased breast cancer mortality in socially isolated patients with ER+ disease [10] may mainly reflect recurrence after hormone therapy is completed. Thus, social isolation may be a major contributor to dormancy, and interventions to reduce its impact of social isolation may be most beneficial if focused on the period following hormone therapy.

The mechanism(s) by which social isolation increases breast cancer mortality in humans remains unclear. Several studies have found tha t loneliness and/or social isolation induce a chronic inflammatory state, and the levels of circulating inflammatory markers are elevated in socially isolated individuals [14,15]. In contrast, socially integrated individuals exhibit reduced inflammatory markers [52]. RNA-Seq analysis of the mammary glands indicated that the top signaling pathways such as the IL6/JAK/STAT3 pathway, which is linked to increased inflammation, were upregulated in socially isolated animals. The IL6/STAT3 pathway is aberrantly hyperactivated in breast cancer [53], is linked to poor prognosis in patients [54], and may drive mammary tumorigenesis in SI rats. Among the inflammatory genes upregulated by social isolation in rats compared to the GH controls were *Csf2* (also known as GM-CSF), *Il18r1*, *Il4r*, and *Lilbr3*. IL4R promotes breast cancer growth [55] and LILRB3 can block antitumor immune activation [56]. CSF2 is linked to the promotion of cancer stem cells via the activation of STAT3 and myeloid-derived suppressor cells [57]. IL18 signaling has been reported to be either pro-tumorigenic or suppressive in tumor development and progression [58].

OXPHOS is another pathway altered by social isolation. In earlier studies, social isolation has been reported to impair the respiratory chain complex, increase the formation of mitochondrial reactive oxygen species (ROS), and cause oxidative damage in various tissues [18], especially in the brain [17]. We found that genes in all four complexes of the respiratory chain, NADH dehydrogenase, succinate dehydrogenase, ubiquinol-cytochrome c oxidoreductase, and cytochrome oxidase were suppressed in SI rats. Furthermore, mitochondrial pyruvate uptake was likely inhibited because both the mitochondrial pyruvate transporter MPC1 and pyruvate dehydrogenase were downregulated in the mammary glands of the SI rats. SI also inhibited the signaling associated with glycolysis, which could reduce the substrates to promote OXPHOS. In addition, several key genes driving the TCA cycle were suppressed in socially isolated rats. While it remains controversial whether reduced OXPHOS increases or inhibits cancer [59], the suppression of OXPHOS is causally linked to diabetes, cardiovascular diseases, and obesity [60,61], which, in turn, are promoted by social isolation.

Previous studies have identified a close association between OXPHOS and inflammation. Oxidative stress induces inflammation [62], which impairs mitochondrial function by suppressing the mitochondrial respiratory chain [63]. It is not clear whether social isolation in our study first induced the inflammation or dysregulation of OXPHOS or whether these activities occurred independently. Impaired OXPHOS is also linked to aging [64] and has been proposed to explain the inflammatory changes related to aging [65]. As aging is a risk factor for social isolation, it is possible that older individuals who are socially isolated develop more age-related physical problems and diseases than socially connected elderly people.

In this study, we did not investigate whether changes in IL6/JAK/STAT3 and OXPHOS signaling explain the increased risk of mammary cancer recurrence in socially isolated rats. In the past, the cancer research field has been focused on identifying a single target that could explain, for example, why cancer progresses and develops resistance to treatments. It is now clear that this approach does not work, but instead, to control cancer, multiple targets need to be considered. Furthermore, cancer is affected by inputs from its environment that include immune, adipose, neuronal, and vascular cells. A recent paper highlights how cancer is affected by body wide influences [13]. Social isolation is an example of influences that are initiated far away from the tumor site, but nevertheless impact tumor growth. Thus, instead of targeting single genes that were differentially expressed in the mammary glands of SI rats, we used the anti-inflammatory herb mix JGT [20,21], which is one of the most widely used traditional herbal mixtures in East Asia. Jaeumganghwa-tang is the Korean name for the mixture, and JGT is called Zi-yin-jiang-huo-tang in China and Jin-koka-to in Japan. In our study, JGT reversed the enrichment of CD14, CSF2, and CXCL10 in the IL6/JAK/STAT3 Hallmark pathway in SI rats. These three genes are linked to the severity of COVID-19 infection [43,44,66]. Others have reported that social isolation impaired the antibody response to the COVID-19 vaccine [67]. Possible effectiveness of Asian herbal mixtures to reduce the symptoms of COVID-19 has been assessed in many studies, and one such mixture (NRICM101) is used in Taiwan for the clinical treatment of COVID-19 [68]. Our study suggests that the anti-inflammatory properties of JGT can prevent social isolation-induced recurrence of mammary tumors in rats.

RAGE, a member of the immunoglobulin superfamily of cell surface molecules, activates NF-κB and increases the production of TGF-β, the pro-inflammatory cytokines IL1 and IL6 [51], and ROS [69]. RAGE is also causally linked to type 2 diabetes [70] and Alzheimer’s disease [71], and social isolation increases the risk of both diseases [3,4]. However, the effect of social isolation on RAGE has not been studied. We found an increase in the RAGE levels in SI rats in normal mammary glands, malignant mammary tumors, and in the brain compared with those in GH rats. However, social isolation did not affect the expression of RAGE ligands *S100a8* and *S100a9*, both of which were inhibited by JGT. JGT did not suppress RAGE expression. These findings suggest that independent pathways drive how social isolation increases RAGE expression and how JGT reduces RAGE ligand expression. Nevertheless, the co-expression of both RAGE and NF-κB supports malignant progression, blocks apoptosis in malignant cells [72], and maintains sustained inflammation [73]. A causal link between RAGE, inflammation, and increased cancer risk, progression, and metastasis has been established [72].

In summary, our results suggest that social isolation may increase breast cancer mortality by allowing the regrowth of dormant mammary tumors and increasing the risk of recurrence after hormone therapy. The increase in recurrence in rats was prevented by the herbal mix JGT, which reversed the social isolation-induced increase in IL6/JAK/STAT3 signaling and the suppression of OXPHOS. How this inflammatory state or metabolic dysfunction occurs in socially isolated individuals remains to be elucidated. In our study, social isolation increased the levels of inflammatory RAGE, which, in addition to activating IL6 signaling [51] might also cause mitochondrial dysfunction [74]. Additional studies are needed to determine whether anti-inflammatory interventions and/or enhancement of mitochondrial metabolism will prevent increased mortality in socially isolated individuals.

## Figures and Tables

**Figure 1 cells-12-00961-f001:**
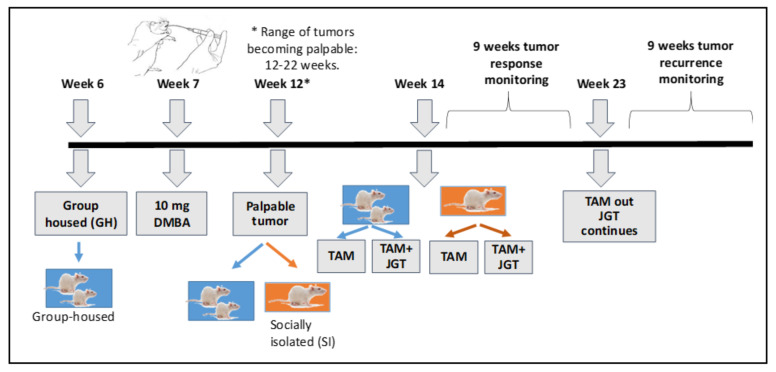
Experimental Design. Estrogen receptor positive (ER+) mammary tumors were induced in 50–day–old Sprague Dawley rats by orally administrating 10 mg of 7,12-dimethylbenz[a]anthracene (DMBA). When a rat developed a palpable mammary tumor, it was either socially isolated (SI) by housing a rat singly or it remained group-housed (GH). Tamoxifen (TAM) or TAM + Jaeumganghwa–tang (JGT) treatment started when the first mammary tumor per rat reached a size of ~11 mm in diameter. TAM was provided at the concentration of 340 ppm via the AIN93G diet and JGT at the dose of 500 mg/kg via drinking water. The tumor responses were assessed, and TAM treatment was stopped when a rat exhibited a sustained complete or partial response for nine weeks. Tumor recurrences after TAM therapy were monitored up to nine additional weeks.

**Figure 2 cells-12-00961-f002:**
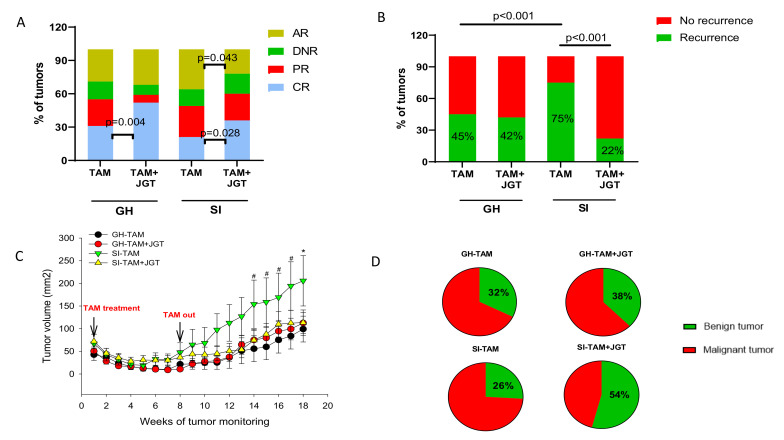
Social isolation increases the risk of local mammary tumor recurrence, and JGT reverts it. (**A**) Percentage of complete responses (CR; blue), partial responses (PR; red), de novo resistant (DNR; green), and acquired resistant mammary tumors (AR; yellow) in group-housed (GH) rats treated with tamoxifen (TAM) (*n* = 58 tumors) or with TAM+ Jaeumganghwa-tang (JGT) (*n* = 44 tumors), and in socially isolated (SI) rats treated with TAM (*n* = 39 tumors) or with TAM + JGT (*n* = 45 tumors). Treatment with JGT increased TAM responsiveness in GH (χ^2^; *p* = 0.004) and SI rats (χ^2^; *p* = 0.03). (**B**) TAM therapy ended after a complete response was maintained for approximately 9 weeks, and then local mammary tumor recurrences (green) were monitored. Social isolation increased recurrence (χ^2^; *p* < 0.001) and JGT reversed the increase (χ^2^; *p* < 0.001). (**C**) After TAM therapy, mammary tumor burden, assessed by measuring the tumor volume (calculated by diameter  ×  width, and sum of all tumor volumes per animal was a tumor burden), was significantly higher in SI rats compared to other groups. Differences were analyzed according to 2-way repeated measures ANOVA followed by the Tukey test. # Indicates statistical significance during weeks when it reached *p*  <  0.05 for GH-TAM vs. SI-TAM. * Indicates statistical significance during weeks when it reached *p*  <  0.05 for GH-TAM vs. SI-TAM and SI-TAM vs. SI-T + J. Means ± SEM of tumor volume at each tumor monitoring week are shown. (**D**) Tumor histopathology showing that JGT significantly increased (χ^2^; *p* < 0.001) the proportion of benign tumors in SI animals from 26% (7 of 27 tumors) to 54% (13 of 24 tumors).

**Figure 3 cells-12-00961-f003:**
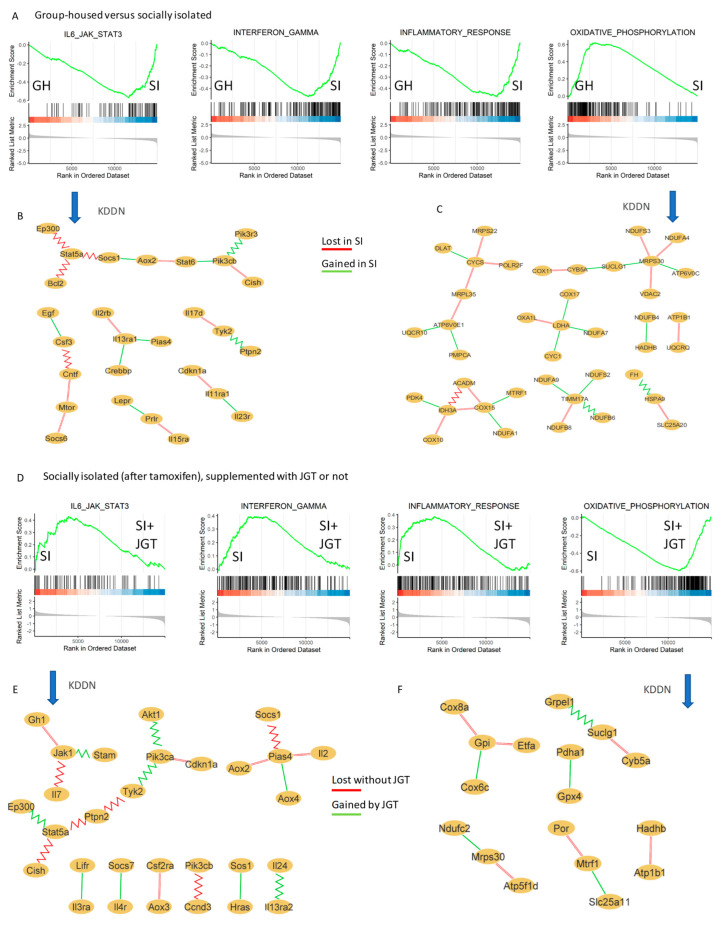
Alterations in the Cancer Hallmark pathways in RNA sequencing analysis in the mammary glands in the group-housed (GH) and socially isolated rats (SI). (**A**) After tamoxifen (TAM) therapy, social isolation increased the IL6/JAK/STAT3, INFγ, and inflammatory response Hallmark pathways and inhibited the oxidative phosphorylation (OXPHOS) pathway, compared with group housing. (**B**,**C**) Knowledge-fused differential dependency network (KDDN) analysis of genes in the IL6/JAK/STAT3 and OXPHOS signaling pathway, respectively, in GH vs. SI rats. Yellow ovals indicate nodes. Red line represents gene–gene interactions that only exist in GH group (lost in SI) and the green line represents gene–gene interactions that only exist in the SI group (gained in SI). (**D**) Adding Jaeumganghwa-tang (JGT) reversed all of these changes in SI rats. (**E**,**F**) KDDN analysis of genes in the IL6/JAK/STAT3 and OXPHOS signaling pathway, respectively, in SI vs. SI + JGT rats. Red line represents the gene–gene interactions that only exist in the SI group (lost without JGT) and the green line represents the gene–gene interactions that only exist in the SI + JGT group (gained by JGT), (**G**) During tamoxifen therapy, JGT had similar effects on the IL6/JAK/STAT3, INFγ, inflammatory response, and OXPHOS Hallmark pathways in GH rats than it did in SI rats after tamoxifen therapy. (**H**,**I**) KDDN analysis of genes in the IL6/JAK/STAT3 and OXPHOS signaling pathway, respectively, in GH without JGT vs. GH + JGT. Red line represents the gene–gene interactions that only exist in the GH group (lost without JGT) and the green line represents the gene–gene interactions that only exist in the GH + JGT group (Gained by JGT). Jagged green and red lines represent the connection also available in the protein interaction database.

**Figure 4 cells-12-00961-f004:**
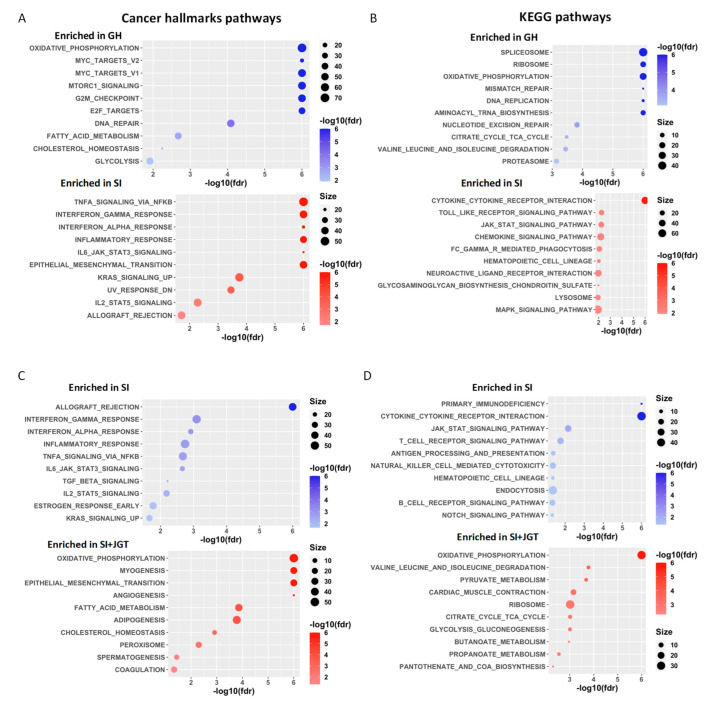
Cancer Hallmark pathways and KEGG pathways enriched in GH and SI treated or not with JGT. Top 10 significantly different Cancer Hallmark and KEGG pathways, generated from RNA-Seq data in mammary glands (**A**,**B**) between the group-housed (GH) and socially isolated (SI) rats after tamoxifen therapy, (**C**,**D**) between SI and SI rats treated with JGT after tamoxifen therapy, and (**E**,**F**) between GH and GH rats treated with JGT during tamoxifen therapy. The color of bubbles represents −log10(fdr). The size of bubbles represents the number of significantly different genes that contributed to the enrichment score.

**Figure 5 cells-12-00961-f005:**
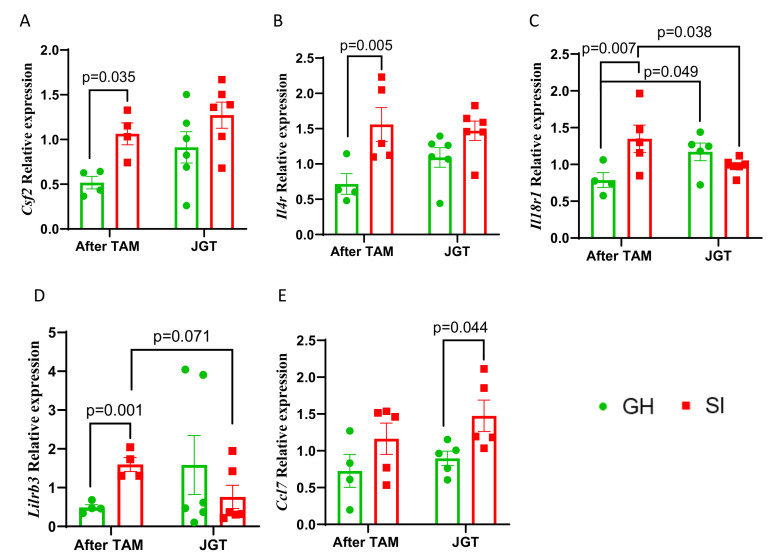
Effect of social isolation (SI) and Jaeumganghwa-tang (JGT) on genes related to inflammatory genes identified as differentially expressed by RNA sequencing analysis in the mammary glands after tamoxifen therapy ended. Expression of (**A**) *Csf2*, (**B**) *Il4r*, (**C**) *Il18r1*, (**D**) *Lilrb3*, and (**E**) *Ccl7* in the mammary glands of group-housed (GH) or SI rats after tamoxifen therapy, and either treated or not treated with JGT. SI significantly increased the expression of *Csf2*, *Il4r*, *Il18r1*, and *Lilrb3*. JGT reverted the expression of *Il18r1* and tended to inhibit *Lilrb3* in SI rats. In GH rats, JGT had the opposite effect on the expression of *Il18r1* than in the SI rats. Differences according to two-way ANOVA followed by the Holm–Sidak test. Means ± SEM, *n* = 4–7 per group are shown.

**Figure 6 cells-12-00961-f006:**
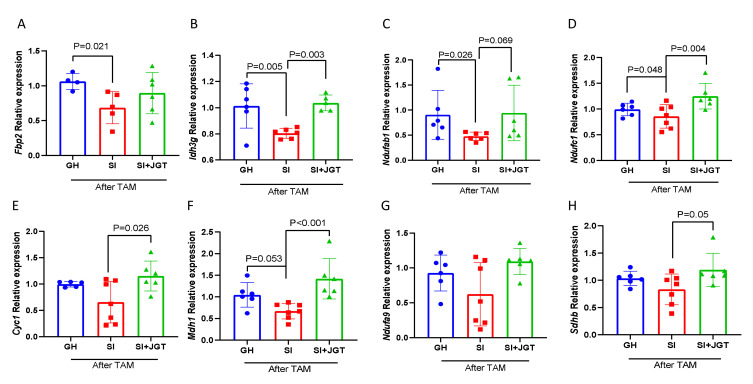
Effect of social isolation (SI) and Jaeumganghwa-tang (JGT) on OXPHOS genes that were suppressed in RNA-Seq analysis in the post-tamoxifen (TAM) mammary glands. Expression of (**A**) *Fbp2*, (**B**) *Idh3g*, (**C**) *Ndufab1*, (**D**) *Ndufc1*, (**E**) *Cyc1*, (**F**) *Mdh1*, (**G**) *Ndufa9*, and (**H**) *Sdhb* in the mammary glands of group-housed (GH) or SI rats after TAM therapy, and either treated or not treated with JGT. SI significantly inhibited the expression of *Fbp2*, *Idh3g*, *Ndufab1*, *Ndufc1*, and non-significantly *Mdh1*. JGT enriched the expression of *Idh3g*, *Ndufc1*, *Mdh1*, and *Sdhb.* Differences according to 2-way ANOVA followed by the Holm–Sidak test. Means ± SEM, *n* = 4–7 per group are shown. Blue circles = GH; red squares = SI; green triangles = SI + JGT.

**Figure 7 cells-12-00961-f007:**
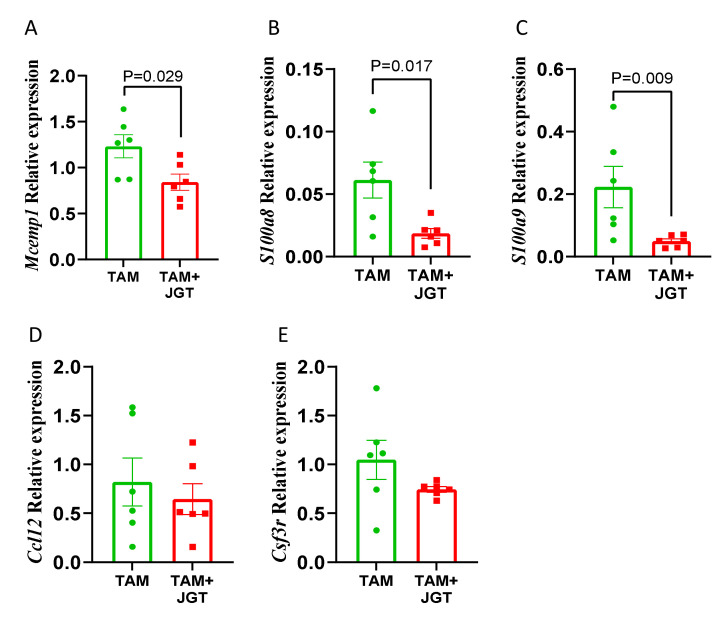
Effect of Jaeumganghwa-tang (JGT) on gene expression in the mammary glands of group-housed (GH) rats during tamoxifen therapy. Expression of (**A**) Mcemp1, (**B**) S100a8, (**C**) S100a9, (**D**) Ccl12, and (**E**) Csf3r in the mammary glands of GH rats during tamoxifen (TAM) therapy, which were either treated or not treated with JGT. JGT significantly decreased the expression of Mcemp1, S100a8, and S100a9. Differences according to the *t* test. Means ± SEM, *n* = 6 per group are shown. Green circles = Tamoxifen treated rats; red squares = tamoxifen + JGT treated rats.

**Figure 8 cells-12-00961-f008:**
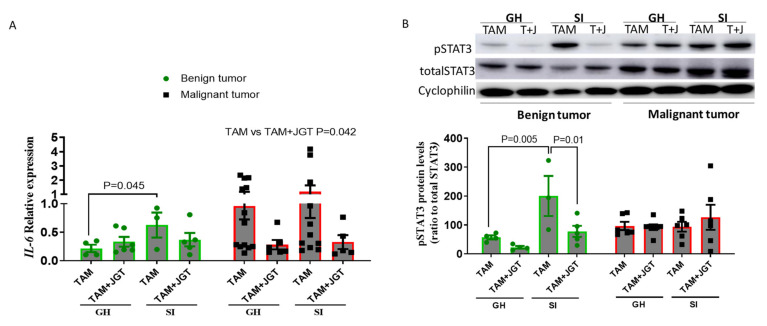
Alterations in the *Il6* mRNA and pSTAT3 protein levels in mammary tumors. (**A**) *Il6* mRNA levels and (**B**) pSTAT3 protein levels in mammary tumors during tamoxifen therapy in GH and SI rats. SI significantly increased the expression of *Il6* in benign tumors, and the difference was eliminated by Jaeumganghwa-tang (JGT). JGT significantly reduced *Il6* in both the GH and SI rats in malignant tumors. In benign tumors, SI significantly increased the protein levels of pSTAT3, and JGT significantly reduced the levels in SI rats. Statistical analysis was conducted using 2-way ANOVA followed by the Holm–Sidak test. Means ± SEM, *n* = 3–13 per group are shown.

**Figure 9 cells-12-00961-f009:**
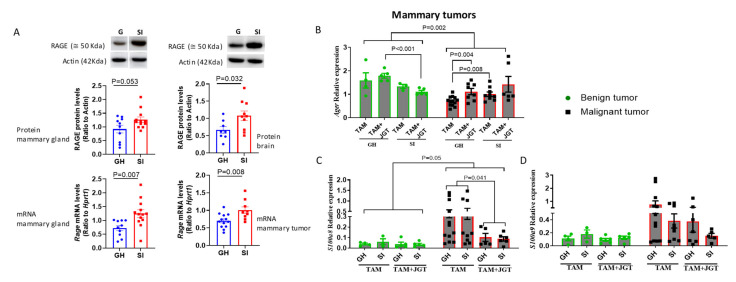
Effect of social isolation (SI) and Jaeumganghwa-tang (JGT) on the receptor for advanced glycation end-products (RAGE). (**A**) Protein and mRNA levels of RAGE/*Rage* in the mammary glands, mammary tumors, and brains of group-housed (GH) and SI rats. The effect of JGT during tamoxifen (TAM) therapy on the gene expression of (**B**) *Rage*, (**C**) *S100a8*, and (**D**) *S100a9* in benign and malignant tumors of GH and SI rats. (**A**) SI increased mRNA expression of *Rage* in mammary glands and tumors and the RAGE protein levels in mammary glands and brain. (**B**–**D**) In benign tumors, JGT reduced the expression of *Rage* in SI rats compared with GH rats. In malignant tumors, JGT increased the expression of *Rage* in GH rats. Expression of *Rage* was significantly lower, and *S100a8* and *S100a9* was higher in malignant tumors compared with benign tumors. JGT treatment decreased the expression of *S100a8* in both the GH and SI rats. Data were analyzed by (**A**) *t*-test, or (**B**–**D**) by 2-way and 3-way ANOVA followed by the Holm–Sidak test. Means ± SEM, *n* = 3–13 per group are shown.

## Data Availability

The data presented in this study are available in https://www.ncbi.nlm.nih.gov/geo/query/acc.cgi?acc=GSE201117 (accessed on 10 April 2022).

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
