# Peer review of "Social Isolation Activates Dormant Mammary Tumors, and Modifies Inflammatory and Mitochondrial Metabolic Pathways in the Rat Mammary Gland"

_cells, 2023, doi:10.3390/cells12060961_

Round 1
Reviewer 1 Report
This is an interesting paper by Andrade and colleagues that builds on their previous studies examining the effects of Jaeumganghwa-tang (JGT) on tamoxifen responsiveness. In the earlier study, the authors show that, while JGT alone is ineffective, when combined with Tamoxifen (Tam), JGT enhances response to anti-estrogen treatment in a rat ER+ tumor model and reversed Tam resistance in a cell line. In this manuscript, the authors perform a similar experiment, but this time they either Group House (GH) or Socially Isolate (SI) the animals. They find that JGT increased the rate of complete responses to tamoxifen, both as observed previously for GH animals and also for SI animals. New here is the observation that SI increased the risk that responding tumors emerge from dormancy and recur and this is more pronounced in SI animals. Their previous study showed that Jaeumkanghwa-tang (JEKHT) reduced the de novo resistance rate (does JGT = JEKHT? unclear in the manuscript). Here they show that JGT prevented the increase in dormant tumor regrowth and local recurrence, and increased the proportion of benign tumors in SI animals. These are intriguing results, although I have questions about dosing and experimental design (see below). To explore molecular mechanism, the authors performed RNA-seq, and use this data to make strong claims about biological function. These claims are largely unbacked by protein analysis, relying heavily on the RNA-seq data and some limited RT-qPCR verification. The molecular analysis as to how JGT may mitigate the enhanced stresses placed on animals in SI is pretty weak. Yet, the authors make some strong conclusions about SI and about the role of JDT. Nevertheless, the data are thought-provoking and if the authors were to shore up their molecular analyses and tone down their conclusions, the manuscript could be acceptable for publication. Below I outline a number of specific concerns.
1) The authors understandably frame this manuscript in terms of human health. Yet, in dosing the animals they use “JGT at a dose of 500 mg/kg was administered via drinking water as described previously (de Oliveira et al., 2019).“ It might be helpful to at least consider putting this number in the context of the human by using body surface area to translate the dose (Reagan-Shaw et al., Faseb, 2007). Although not perfect, this calculation is more meaningful and would give the reader some sense of whether a person could consume sufficient JGT to obtain the observed effects.
I am unclear about the experimental design. Was the experiment performed once? And, when it was performed, there were four groups: group-housed treated with tamoxifen (n=19), group-housed treated with tamoxifen+JGT (n=17), socially isolated treated with tamoxifen (n=17) and socially isolated treated with tamoxifen+JGT (n=18). If this is the case, in my view, the group housed animals represent an n=1 if they were all housed together (as n=19 +T and n=17 +T+JGT), but maybe they were housed in subgroups? For group housed animals, what does an “n” represent. The authors need to be clear on this point because it is important in considering the RNA-seq analysis and RT-qPCR verification (how many “n”s went into these studies). Because if 5 animals are housed together, I think they represent 1 “n” (a cage = 1n), not that each one of those 5 animals om a cage is an “n” (a cage = 5n).
Figure 2 shows the results from the animal studies. Most impressive is the difference between the recurrence, the size and the malignancy of tumors under Tam versus Tam+JGT treatments in SI conditions. Why did the authors used the Holm-Sidal test following the ANOVA analysis rather than Tukey?
Figure 3. The legend explains the panels out of order, which is confusing. The enrichment plots are blurry and difficult to read. Does every blue arrow represent translation by KDDN because only the arrow on the left is labeled. In E, F, H, and I, I don’t see double lined red connections, only single red lines. I also see jagged green and jagged red lines that have no explanation. The gene names in the yellow ovals are difficult to read. The notion of “lost” and “gained” is confusing. Table 1 is a very reader unfriendly way to display KEGG results; the authors should make an effort to graph the data.
Figures 4-6: I agree with the authors that “it is customary to verify the validity of key results from RNA-seq and other omics analyses ” and, indeed, the validation of a smattering of genes in identified pathways is presented. But, RT-qPCR only tells us whether the changes identified in the RNA-seq data can be reproduced using a different method. It does not tell us whether the changes in gene expression have functional consequences. For this, the authors need to measure the inflammatory response and OXPHOS (e.g. Acin-Perez et al., EMBO, 2020) using immunohistochemical or Western blot markers, commercially available kits or other methods. They did this for RAGE in Figure 8. Without this additional validation, the authors haven’t demonstrated that the changes observed in gene expression result in changes in protein levels (and therefore presumably in physiological changes) in the animals.
In Figure 4, the data that JGT does anything is weak: 1 gene of 5 significantly changed and a very modest change at that. It is odd because gene expression in seems to change with JGT treatment in both GH and SI animals. I gather that only significant changes are called out but the data are very variable and would benefit from additional analysis “n”s. Again, I am wondering about the housing (see above). Does each dot represent one animal that was housed with all the other “n”s? or does it represent the average of all those “n”s in the cage. Because to me, housing animals together means they equal 1n.
I am also confused by why Figure 4 and 5 are presented differently. In Fig 4 we see GH/SI After Tam and (I think) After Tam+SI (although the labels on the graphs are not clear). In Figure 5, we aren’t shown GH+JGT. Is this because there are the largest changes in gene expression with GH+JGT? Again, these data are variable and show modest changes, but are used to make major claims in the Discussion.
Figure 7. The authors look at Il6 mRNA by RT-qPCR and pSTAT3 protein levels in mammary tumors, which is confusing because they previously took care to explain why they performed RNAseq on mammary glands rather than tumors. “We used RNA-seq analysis to determine which signaling pathways were altered by social isolation and JGT treatment. We considered whether comparisons should be performed in mammary glands or tumors……Hence, the fourth abdominal mammary gland was obtained from GH or SI rats for RNA-seq. ” So here and in Figure 8 they switch gears and analyze benign versus malignant tumors without explaining why. It would be appropriate to include a better transition sentence that explain the rationale to the reader than “We also determined the changes in IL6 and STAT3 levels in mammary tumors. ”
The individual data points aren’t shown on the graphs as they are in Figures 4-6. It is confusing that the order of the samples in the Western blot and the graphs don’t match. The authors say “JGT reduced Il6 and pSTAT3 expression in mammary tumors; the reduction was observed in malignant tumors for Il6 and in benign tumors for pSTAT3. ” But the statistical analysis is confusing. For Il3 in malignant tumors, is the p=0.042 value exactly the same for GH/TAM compared to GH/TAM+JGT and for SI/TAM compared to SI/TAM+JGT? I don’t understand the grouping. For STAT3, it appears that only benign SI tumors (p=0.01) were analyzed; there is no statistical analysis of GH tumors. And without a loading control for these Western blots, the authors can only say that pSTAT3 expression changed relative to total STAT. Bottom line: what’s the point of this analysis? Why these two disparate endpoints: one mRNA and one phosphoprotein? What does it tell us about benign versus malignant tumors, SI and JGT? This section is confusing, and breaks the flow.
Figure 8. The authors look at RAGE by RT-qPCR and Western blot. Again, the authors should show individual data points. For panel A, why don’t we see what happens to RAGE protein after JGT treatment? In panel B we see Ager mRNA but as the authors say “Paradoxically, benign tumors in GH rats exhibited higher Rage mRNA levels than malignant tumors did. JGT did not affect RAGE levels (Fig. 8B). ” I am confused again about the statistical comparison (p=0.041 in panel C. We do see that JGT suppresses RAGE ligand expression but in both GH and SI conditions. Also is this only statistical for S100a8? I am not sure what to make of these data.
As a consequence of confusions in the data, I think the discussion overreaches in making conclusions in a number of places.
1) “Results showed that hormone therapy-induced, mitochondrial-defective (OXPHOS down) dormant breast cancer cell population exited dormancy via an IL6/Stat3/Notch3-mediated induction of mitochondrial activity (Sansone et al., 2016) suggesting that inflammation may drive recurrence.” The authors do not show changes in Notch3 or mitochondrial activity and the only way to show that inflammation drives recurrence is to prevent the inflammation and have no occurrence. JGT might be a drug that could do this because it inhibits inflammatory pathways. Yet, JGT alone in GH control animals did not inhibit mammary tumor recurrence, which is consistent with our earlier study showing that JGT monotherapy did not prevent mammary tumor growth (de Oliveira et al., 2019). So, I don’t think the authors can make this assertion.
2) “Our study suggests that social isolation may contribute to the severity of Covid-19 infection.” I don’t think one can analogize the change in the expression of one inflammatory gene in this SI paradigm to understanding how Covid-19-mandated SI impacts the severity of Covid infection.
3) “The increase in recurrence in rats was prevented by the herbal mix JGT, which reversed the social isolation-induced increase in IL6/JAK/STAT3 signaling and suppression of OXPHOS.” The effects of JGT on tumor recurrence were much stronger than any effects observed molecularly. The strongest effects in the paper had to do with SI. I think the authors will want to carefully consider the conclusions that make about the molecular effects of JGT.
Reviewer 2 Report
The manuscript by Dr. de Oliveira et al reports experiments on the effects of a herbal mixture "JGT" on the recurrence of DMBA-induced tumors after tamoxifen treatment of Sprague-Dawley rats. The authors have previously described the model and reported the sensitizing effect of JGT on the tamoxifen responsiveness of DMBA tumors. Here they apply the model and experimental setup to investigate the impact of social isolation vs conventional group housing of the DMBA tumor-bearing rats. They demonstrate that JGT decreases the reappearance of tumors after tamoxifen treatment which effect is associated with decreased inflammatory mediators and increased parameters of oxidative metabolism.
These results are interesting and important and deserve further investigation also considering human breast cancer. However, very wide questions are asked (social isolation vs integration and the application of a herbal mixture which, although widely used and highly interesting, has obviously not yet been thoroughly analyzed) which should be pointed out considering potential further studies and applications.
The authors know well the rather complex and laborious experimental model that they use. Some practical questions arise. Was the proportion of benign tumors that appeared after tamoxifen treatment determined by histopathological analysis only? What did the morphology represent and was it similar in all "benign" tumors? Were all tumors (benign and malignant) included in RNAseq?
The results from RNAseq experiments and analyses support the conclusions. However, they could be further elaborated to make the presentation clear and more reader-friendly.
Round 2
Reviewer 1 Report
The authors have improved the manuscript and addressed many (but not all) of my comments. To be clear, I still find some of their assertions simply unscientific. For example, they say "Changes in IL6 mRNA levels is sufficient to indicate that IL6 signaling is altered.” No, I don’t think so. As researchers, we don’t look at mRNA levels to see the level of the protein. We do Elisa assays or immunoblotting. But I get it, you did this experiment once and maybe you didn’t harvest what you need to answer my very reasonable question or maybe you simply don’t want to put that effort in. I don't know, but I don’t think it is appropriate for the authors to assert new rules about what is considered acceptable evidence. In this way the paper has weaknesses, but I find the improvements have made the manuscript acceptable for publication.
Author Response
We agree with the reviewer that our response to the question relating to IL6 and pSTAT3 was too simplistic and a target for misinterpretation. As the reviewer points out, protein levels are more informative regarding activation (or suppression) of a gene. However, we attempted to justify in our response that in our study the Il-6 mRNA was sufficient to determine, since when IL-6 protein is secreted, one of its down-stream targets – STAT3 – should be activated. Since pSTAT3 was upregulated, we took this to mean that mRNA upregulation of IL6 was indicative that IL6 protein levels were increased and IL6 was activated. Therefore, we believe that it is reasonable to infer that IL-6 signaling is activated when pSTAT3 is elevated and IL6 mRNA levels are increased.